# Onboard Real-Time Hyperspectral Image Processing System Design for Unmanned Aerial Vehicles

**DOI:** 10.3390/s25154822

**Published:** 2025-08-05

**Authors:** Ruifan Yang, Min Huang, Wenhao Zhao, Zixuan Zhang, Yan Sun, Lulu Qian, Zhanchao Wang

**Affiliations:** 1Aerospace Information Research Institute, Chinese Academy of Sciences, Beijing 100094, China; yangruifan23@mails.ucas.ac.cn (R.Y.); huangmin@aoe.ac.cn (M.H.); zhaowh@aircas.ac.cn (W.Z.); zhangzixuan@aircas.ac.cn (Z.Z.); sunyan@aircas.ac.cn (Y.S.); qianll@aircas.ac.cn (L.Q.); 2School of Optoelectronics, University of Chinese Academy of Sciences, Beijing 100094, China; 3Key Laboratory of Computational Optical Imaging Technology, Chinese Academy of Sciences, Beijing 100094, China

**Keywords:** hyperspectral, onboard, real-time processing, FPGA-ARM, embedded system

## Abstract

This study proposes and implements a dual-processor FPGA-ARM architecture to resolve the critical contradiction between massive data volumes and real-time processing demands in UAV-borne hyperspectral imaging. The integrated system incorporates a shortwave infrared hyperspectral camera, IMU, control module, heterogeneous computing core, and SATA SSD storage. Through hardware-level task partitioning—utilizing FPGA for high-speed data buffering and ARM for core computational processing—it achieves a real-time end-to-end acquisition–storage–processing–display pipeline. The compact integrated device exhibits a total weight of merely 6 kg and power consumption of 40 W, suitable for airborne platforms. Experimental validation confirms the system’s capability to store over 200 frames per second (at 640 × 270 resolution, matching the camera’s maximum frame rate), quick-look imaging capability, and demonstrated real-time processing efficacy via relative radio-metric correction tasks (processing 5000 image frames within 1000 ms). This framework provides an effective technical solution to address hyperspectral data processing bottlenecks more efficiently on UAV platforms for dynamic scenario applications. Future work includes actual flight deployment to verify performance in operational environments.

## 1. Introduction

Hyperspectral imaging (HSI) technology, emerging in the 1980s, has become a cornerstone tool in remote sensing due to its unique advantage of “integration of spatial and spectral information” [1,2]. This capability allows for the simultaneous acquisition of spatial imagery and high-resolution spectral information across hundreds of contiguous narrow bands. Renowned for its powerful capacity for ground object identification and inversion, HSI has found extensive practical application in environmental monitoring, precision agriculture, geological exploration, disaster emergency response, and other domains, demonstrating significant practical value [3,4,5,6].

In recent years, the maturation of Unmanned Aerial Vehicle (UAV) technology has provided a revolutionary platform for hyperspectral imaging [7,8,9]. Compared to traditional platforms like satellites and manned aircraft, deploying HSI on UAVs offers distinct advantages: lower cost, flexible low-altitude flight capabilities, and rapid response times. These advantages have substantially expanded the application boundaries of hyperspectral technology [10,11,12,13,14,15].

However, the practical implementation of UAV-borne hyperspectral imaging still faces a critical bottleneck: the pronounced conflict between the “massive volume” of hyperspectral data and the requirement for “real-time processing”. On one hand, hyperspectral imaging necessitates the simultaneous recording of two-dimensional spatial information and spectral dimension information, resulting in substantial data volumes [16]. Hyperspectral cameras can generate hundreds of megabytes (MB) of data per second. For example, using the specifications from this study’s experimental setup—a spatial resolution of 640 × 512 pixels, 270 spectral bands, and 2 bytes per pixel—the size of a single hyperspectral image already reaches 168.75 MB. On the other hand, constrained by the limited computational resources of airborne platforms, traditional hyperspectral data processing relies on a sequential “fly-acquire-download-offline processing” paradigm [17]. Data must first be stored on the airborne device, transmitted back to the ground station, and then imported into high-performance computers for post-processing tasks like image preprocessing and feature extraction. This results in significant latency between data acquisition and the final output of image results. Such delays not only fail to meet the real-time decision-making demands of various dynamic scenarios but also hinder the practical deployment of many spectral payloads currently confined to laboratory settings.

Therefore, developing real-time processing technology for UAV-borne hyperspectral imaging has become crucial for enabling its large-scale application. A real-time processing system must perform high-speed storage and preliminary analysis concurrently with data acquisition, ensuring users can access critical information (such as real-time quick-look images) during the flight itself to support on-site decision-making. Although existing research has attempted to enhance processing speed through algorithm optimization or hardware acceleration [18,19,20,21], there remains a lack of systematic, real-time processing solution designs tailored to the constraints of UAV platforms, such as miniaturization, low power consumption, and vibration resistance. These solutions struggle to adapt to the resource limitations inherent in UAV platforms.

To address the aforementioned challenges, this paper designs a real-time processing system for UAV-borne hyperspectral imaging. The principal contributions of this work are summarized as follows:Proposal and implementation of a comprehensive system-level airborne hyperspectral real-time processing solution: Addressing the lack of integrated system approaches in existing research, this study pioneers a full-chain real-time processing system unifying data acquisition, high-speed storage, real-time computation, and quick-look visualization. This achievement bridges the critical technology gap between UAV platforms and operational hyperspectral applications.Innovative adoption of Field-Programmable Gate Array-Advanced RISC Machines (FPGA-ARM) dual-processor architecture with hardware–software co-optimization: A heterogeneous computing platform was designed leveraging FPGA-ARM collaborative processing, where FPGA handles data offloading and buffering while ARM executes storage and processing tasks, fully exploiting hardware parallelism. Concurrently, key software optimizations—including a multithreaded concurrency model, batched writing strategy, asynchronous file I/O, and reliability-enhanced communication protocols—were developed for ARM to resolve resource contention and real-time bottlenecks, significantly elevating system throughput to 200 frames per second (at 640 × 270 resolution, matching the camera’s maximum frame rate).Experimental validation successfully confirmed the feasibility and stability of the system: (1) We tested data acquisition and storage capabilities, verifying reliability of dual-transmission schemes based on UDP; (2) under simulated airborne push-broom imaging conditions using an optical swing simulator, the system demonstrated image acquisition capabilities and real-time preview functionality via HDMI OUT interface; (3) using relative radiometric correction as a representative task, the system’s real-time processing capacity was validated.

The second part of this article details the design of the airborne hyperspectral image real-time processing system. The third part of this article focuses on presenting the experimental results. Finally, the fourth part provides conclusions and discussion.

## 2. Onboard Real-Time HSI Processing System

### 2.1. System Overview

The system architecture and operational principles are illustrated in Figure 1.

The UAV-borne real-time hyperspectral imaging processing system comprises the following components: a dual-processor FPGA-ARM architecture-based data acquisition and processing module serving as the computational core, a key control module, a hyperspectral camera, an IMU sensor for spatial positioning and attitude determination, a Serial Advanced Technology Attachment Solid State Drive (SATA SSD) for data storage, and a power module.

The system workflow operates as follows: The power module supplies 12 V power to all other components. Within the FPGA processor, an integrated level shifting module provides regulated voltages to submodules while ensuring electrical safety. The ARM processor incorporates a 12 V-to-5 V converter to power the SATA SSD. Communication between the key control module and the FPGA master chip utilizes RS422 serial protocol. Upon system power-up, the operator can use the push-button interface to transmit control commands—including image acquisition frame rate, camera exposure time, and gain mode—to the FPGA. The FPGA subsequently relays these parameters to both the hyperspectral camera and the IMU sensor.

The FPGA receives hyperspectral image data via the CameraLink interface, buffers it, and transmits it to the ARM subsystem through User Datagram Protocol (UDP). Concurrently, the FPGA acquires IMU navigation data through a UART serial interface, buffers the incoming stream, and relays it to the ARM via serial transmission. The ARM subsystem handles image reception and real-time processing, ultimately storing results on the SATA SSD while enabling quick-look visualization through its HDMI OUT port.

Section 2.2 details the hyperspectral camera specifications, followed by Section 2.3 describing the IMU sensor characteristics. Section 2.4 documents the SATA SSD storage subsystem, while Section 2.5 elaborates on the push-button control module. Section 2.6 presents the Data Acquisition and Processing Module, culminating in Section 2.7 with the integrated hardware architecture design.

### 2.2. Hyperspectral Camera

The hyperspectral camera employed in this study is the Micro-Hyperspec SWIR (Short-Wave Infrared) 640 model [22] manufactured by Headwall Photonics. This camera features a spectral range of 900–2500 nm and utilizes a Stirling-cooled Mercury Cadmium Telluride (MCT) sensor, with a total weight under 2 kg. Data acquisition is facilitated through a Base CameraLink interface. The optical dispersing element employs an Offner convex holographic reflection grating, and the internal light path contains no refractive optics, completely eliminating chromatic aberration while minimizing stray light to enhance image quality. This camera configuration provides 640 spatial pixels × 270 spectral bands. Operating in push-broom imaging mode, each acquisition captures 640 spatial pixels along the *x*-axis with full spectral resolution (270 bands), while the UAV’s motion along the *y*-axis enables construction of the hyperspectral data cube.

Figure 2 presents a 3D rendered exterior view of the camera, with Table 1 listing its technical specifications.

### 2.3. Inertial Measurement Unit Sensor

The IMU sensor employed in this study is the EPSILON D-series [23] miniature inertial Real-time kinematic (RTK) Global Navigation Satellite System (GNSS) integrated navigation system manufactured by FDISYSEMS. This sensor incorporates dual independent sets of triaxial gyroscopes, dual triaxial accelerometers, a high-performance triaxial magnetometer array, barometric pressure sensor, and thermometer. Its integrated dual-antenna differential GNSS delivers centimeter-level positioning accuracy and dual-antenna heading determination. The unit features a robust Sigma-Point Kalman Filter (SPKF) (FDIsystems Ltd., Hefei, China) and tight-coupling navigation algorithm, achieving sensor sampling rates up to 1000 Hz with inherent magnetic interference immunity. The EPSILON D IMU is exceptionally suited for SWaP-constrained UAV platforms requiring high-precision navigation.

Figure 3 presents the physical configuration of the EPSILON D IMU sensor, with Table 2 detailing its technical specifications.

### 2.4. Serial Advanced Technology Attachment Solid State Drive

The storage subsystem employs a SAMSUNG 870 EVO 2.5-inch SATA SSD [24] with 2TB capacity. In this study’s implementation, each image frame occupies 640 × 270 × 2 = 345,600 bytes (representing 640 spatial pixels × 270 spectral bands × 16-bit depth). At 100 fps acquisition rate, the sustained data throughput reaches 32.96 MB/s. Consequently, the 2 TB capacity supports 17.67 h of continuous operation—sufficient for extended UAV missions. This SSD delivers rated sequential read/write speeds exceeding 500 MB/s, comfortably satisfying the 300 MB/s storage bandwidth requirement. Additionally, its compact form factor and lightweight design make it ideally suited for SWaP-constrained airborne platforms.

Figure 4 illustrates the physical configuration of the SATA SSD, while Table 3 details its technical specifications.

### 2.5. Key Control Module

The key control module configures the hyperspectral camera’s gain mode, acquisition cycle, and exposure time. It communicates with the FPGA host controller via RS422 serial interface to transmit current parameter settings. This module comprises a key-digit interface panel and a core control board. The implemented key-digit panel integrates a 4-digit 8-segment LED display and four tactile keys. The display dynamically presents different parameter values, with key 1 designated for mode selection, key 2 and 3 for value increment and decrement, and key 4 for parameter confirmation.

Figure 5 shows the appearance of key-digit interface panel while Figure 6 shows the core control board. Table 4 provides a comprehensive display mapping reference table.

### 2.6. Data Acquisition and Processing Module

As illustrated in Figure 1, the Data Acquisition and Processing Module comprises an FPGA processor and an ARM processor. Consistent with the system workflow described in Section 2.1, the FPGA primarily handles offloading and buffering of hyperspectral data and IMU navigation data. The ARM processor undertakes the most computationally intensive tasks of data storage and image processing.

The FPGA processor receives image data from the hyperspectral camera via the CameraLink interface, employing DDR3 SDRAM (Advanced Micro Devices, Inc., San Diego, CA, USA) as a high-speed buffer for data reception and temporary storage. After internal buffering, the image data and ancillary information are transmitted to the ARM subsystem via Ethernet using UDP. Simultaneously, the FPGA acquires IMU navigation data through RS422 serial communication. This inertial data is buffered and then forwarded to the ARM via asynchronous serial transmission at 3.3 V TTL levels.

The ARM subsystem receives hyperspectral data through its Gigabit Ethernet interface and IMU navigation data via its UART port. Internally, the ARM executes our novel algorithm software that bypasses system caching to directly write raw data to the SATA SSD, achieving high-speed storage and completing the data transmission pipeline. Following real-time processing of hyperspectral data, the ARM outputs processed imagery through its HDMI OUT port for quick-look visualization.

The FPGA-ARM dual-processor architecture overcomes single-processor performance limitations through heterogeneous task partitioning. Specifically, the FPGA operates as a hardware acceleration engine dedicated to high-speed hyperspectral data acquisition, buffering, and real-time preprocessing. This configuration effectively frees ARM computational resources for core processing tasks. Meanwhile, the ARM leverages its multicore architecture and GPU acceleration capabilities to focus on complex algorithm execution. This parallel collaboration significantly enhances real-time processing efficiency. Furthermore, the architecture delivers high reliability—the FPGA’s hardwired critical logic mitigates software runtime risks, while the ARM supports dynamic module loading for enhanced fault tolerance. Its modular design facilitates future functional expansion, providing flexible support for stable operation and technological iteration in complex scenarios.

#### 2.6.1. Hardware Specifications

Supplementary hardware specifications for the FPGA and ARM are detailed below:

The selected Xilinx Kintex-7 FPGA (XC7K325T) [25] is a critical enabler for high-performance, SWaP-optimized (Size, Weight and Power-optimized) UAV applications, with key adaptations including: ① Abundant logic resources: Over 320,000 logic cells and extensive DSP slices deliver sufficient parallel processing capacity for real-time data stream requirements; ② Substantial high-speed on-chip memory: Large-capacity Block RAM provides an optimal buffer for hyperspectral image data; ③ High-speed serial transceivers: Integrated multi-channel GTX transceivers offer native physical-layer support for Gigabit Ethernet PHY (enabling UDP output) and CameraLink deserialization; ④ Versatile I/O interfaces; ⑤ Industrial temperature range (−40 °C to +100 °C); ⑥ High operating frequency.

The ARM master controller employs the Rockchip RK3588 SoC [26], whose heterogeneous computing power and efficient I/O subsystem are pivotal for high-speed hyperspectral data acquisition and storage. Core computational advantages include: ① Flagship CPU/GPU/NPU configuration: Integration of quad-core Cortex-A76, quad-core Cortex-A55, ARM Mali-G610 MP4 GPU, and a 6-TOPS NPU; ② High-efficiency memory subsystem and I/O bandwidth: Support for dual-channel LPDDR4X/LPDDR5 memory ensures low-latency transfer of massive hyperspectral data streams between CPU and I/O controllers (particularly SATA), effectively mitigating memory bandwidth bottlenecks; ③ Native hardware support for SATA 3.0 interfaces, enabling direct drive of SATA SSD at maximum performance; ④ Native integration of Gigabit Ethernet and multiple high-speed UARTs, perfectly aligning with the system’s data input (UDP, UART) and storage (SATA) requirements; ⑤ Onboard HDMI OUT port supporting HDMI 2.0 for real-time display of processed quick-look imagery.

#### 2.6.2. ARM-Side Software Implementation

To address real-time bottlenecks and resource contention in hyperspectral data reception and storage, this study implements multi-dimensional optimizations within the ARM-side real-time data processing pipeline, focusing on four core technical approaches: multithreaded concurrency, I/O efficiency enhancement, memory management refinement, and computational resource scheduling, as detailed below:

First, as illustrated in Figure 7, to achieve parallel execution of data reception, storage, and IMU processing, the software employs a three-thread concurrent model: The reception thread handles UDP hyperspectral data retrieval and integrity verification; the write thread exclusively persists buffered data to the SATA SSD; and the IMU thread independently processes navigation data from UART and synchronously writes to storage. These dedicated roles eliminate mutual blocking between reception and storage operations inherent in single-threaded serial execution, significantly boosting system throughput.

Thread thrashing was eliminated through CPU affinity binding and thread priority scheduling. The ARM processor features eight CPU cores (0–7), with Cores 0–3 optimized for background data processing and Cores 4–7 dedicated to rendering tasks. Given its critical role in preventing packet loss during high-frequency data reception, the data receiving thread was bound to Core 3 (typically the most idle) with the highest priority. Since the writing thread completes faster than the receiving thread within the same cycle and exhibits higher temporal tolerance, it was assigned to Core 2 (moderately idle) with lower priority. The IMU processing thread, requiring minimal computational effort due to small data volumes, was bound to Core 1 with the lowest priority, while Core 0 handles essential system operations.

Second, mitigating performance degradation from frequent I/O operations during high-frequency writes, the software adopts batched writing and asynchronous file I/O mechanisms: The reception thread accumulates packets until reaching a preset raw file size (e.g., 512 frames) before bulk-transferring data to the write thread. This enables overlapping of data transfer and CPU computation, further reducing I/O wait times and minimizing overall processing latency.

We implemented a large-capacity buffer for UDP Socket data caching, providing enhanced fault tolerance throughout the data reception pipeline. Complementing this, a ping-pong buffering mechanism (dual-buffer pool) was deployed. Two pre-allocated buffers operate alternatively: while the receiving thread populates Buffer A, the writing thread simultaneously processes Buffer B, achieving zero-wait switching.

To eliminate unnecessary variable space overhead and further accelerate processing, we extensively applied zero-copy techniques during programming. When notifying the writing thread to transfer received buffer data to the SSD, the system directly transfers content via memory-mapped operations. This bypasses redundant data copying between kernel-space and user-space.

To enhance communication reliability, we implemented dedicated protocols for FPGA-ARM data transfer. Each UDP packet containing hyperspectral image data incorporates sequential frame and packet numbering. The ARM processing pipeline verifies data continuity by monitoring these identifiers to detect packet loss. For serial-transmitted IMU data, each message features a fixed 2-byte header and a checksum byte, enabling error detection and positional resynchronization. Crucially, both hyperspectral ancillary data and IMU messages contain precisely aligned timestamps, permitting cross-validation for temporal synchronization integrity.

Figure 8 illustrates the program flow of the hyperspectral data receiver thread. Each hyperspectral image frame is fragmented into 271 packets for transmission and reception. Upon receiving each packet, validation is performed based on its frame number and packet number.

### 2.7. Integrated Hardware Architecture Design

Figure 9 showcases the physical prototype of the complete system. Figure 10 reveals the internal view of the dual-layer physical assembly. Considering UAV payload constraints, the entire assembly adopts a compact and lightweight structural design. Internally, components are organized in a dual-layer vertical arrangement: the upper layer houses the secured Data Acquisition and Processing Unit, while the lower layer contains the mounted hyperspectral camera and the IMU sensor.

The integrated system combines multiple functions including hyperspectral imaging, IMU inertial navigation, and image acquisition and storage. Despite this integration, it maintains a compact and miniature size, with a total weight optimized to be lightweight, controlled at approximately 6 kg. This enables it to meet the payload requirements of industrial small-to-medium-sized drones.

Based on component product specifications [22,23,24,25,26], the camera has a maximum power consumption of 14 W, the IMU 3 W, the SSD 4.5 W, the FPGA 9 W, and the ARM subsystem 8 W. This yields a theoretical system power estimate of approximately 38.5 W, excluding minor auxiliary modules. During actual testing with a 12 V external power supply, the device stabilized at 2.5 A current draw. We therefore calculate the operational power consumption as 40 W (12 V × 3.3 A), which aligns closely with the theoretical estimation.

Consistent with component specifications [22,23,24,25,26], the system’s industrial-grade components ensure broad operational temperature coverage from −40 °C to 60 °C, satisfying requirements for most UAV deployment scenarios. Regarding vibration stability, the apparatus is mounted with shock-absorbing dampeners during installation. When deployed on UAVs, it will be further isolated using gimbal stabilization systems, thereby guaranteeing vibration resistance.

As the system targets low-altitude UAV operations—classified as routine scenarios—magnetic field interference and other complex environmental factors fall outside this study’s scope.

Table 5 lists the overall system parameters.

## 3. Results

### 3.1. Data Acquisition and Storage Function Experiment

Figure 11 illustrates the complete data transmission pipeline of the system. Theoretically, the hyperspectral data transmission follows this serialized pathway: raw data acquisition from the hyperspectral camera to the FPGA; FPGA-side data offloading and buffering; ARM-side image processing; and final storage to the SATA SSD via ARM. Each segment maintains bandwidth exceeding 1000 Mb/s, making the ARM-side processing speed the primary bottleneck for overall throughput.

Each camera frame contains 640 spatial pixels × 270 spectral bands with 2-byte pixels, resulting in a per-frame size of 640 × 270 × 2 × 8 = 2,764,800 bits. At 100 fps, this corresponds to a theoretical acquisition/storage rate of 2,764,800 × 100 ÷ 1024 ÷ 1024 ≈ 263.67 Mb/s.

To validate the system’s capability for rapid airborne hyperspectral data acquisition and storage, we conducted practical throughput tests: the system executed nine experimental trials (three repetitions each at 50, 100, and 200 fps) with 30 min continuous operation per trial. During transmission, each frame was fragmented into 270 UDP packets. Packet loss was categorized into individual UDP packet loss and complete frame loss. We counted the number of data losses when using UDP transmission and observed when these situations would occur.

Table 6 documents the experimental packet loss statistics. At 50 fps, the system exhibited zero packet loss. Under 100 fps operation—the primary frame rate for practical applications—average packet loss remained minimal at approximately 2 incidents per 30-min session. When pushed to 200 fps (near the camera’s maximum capability), packet loss averaged 6 occurrences per half-hour trial.

Given the push-broom imaging paradigm where each frame comprises 270 UDP packets—with individual packets corresponding to single spectral channels per spatial row—isolated packet loss has negligible operational impact. Complete frame loss—where all 270 packets of a frame are missing—produces discernible artifacts and represents an intolerable failure condition. Our measurements confirm the Complete Frame Loss Rate remains ≤ 0.06‱. This occurrence rate falls within acceptable operational limits for mission-critical applications.

Here, we provide supplementary clarification regarding packet loss tolerance. For the adopted push-broom imaging mode, each image frame corresponds to a complete spatial line scan (640 pixels) transmitted as 270 UDP packets with each packet representing spectral data for one pixel position along the line. Crucially, the loss of a single UDP packet results only in the loss of spectral data for one specific pixel position within that frame (line scan). This partial loss is generally tolerable for downstream processing. Conversely, the complete loss of all 270 packets associated with a single frame (an entire spatial line) constitutes a significant and often unacceptable error for mission-critical analysis. Our experimental validation demonstrates sustained transmission exceeding 200 fps for 640 × 270 pixel images during continuous 30 min operation, where complete frame loss was exceptionally rare with only 0–2 occurrences. This corresponds to a complete frame loss rate of <0.06‱, meeting the stringent requirements of the vast majority of envisioned real-time UAV hyperspectral applications.

Our observations indicate that packet and frame losses predominantly occur during ARM-side data relay and bulk storage operations. When the write thread performs large-file writes, the memory controller bandwidth becomes saturated, causing receiver-side DMA operations to experience delayed memory access authorization. This results in receive buffer overflows. Based on this analysis, further reducing the loss rate could be achieved by decreasing the frequency of file writes. For instance, the current approach of writing every 512 frames could be adjusted to writing at higher intervals, such as every 1024 frames or more per write operation.

Conclusively, experimental evaluations within this study validate the hyperspectral data transmission scheme based on UDP protocol, achieving a sustained storage rate exceeding 200 frames per second for 640 × 270-pixel images (the maximum frame rate supported by the employed camera) while maintaining a data loss rate below 0.01%. This performance level effectively fulfills the demanding requirements of most onboard hyperspectral real-time processing scenarios. The UDP-based solution further distinguishes itself through advantageous characteristics such as minimal hardware requirements, low implementation cost, strong scalability, and high operational flexibility, establishing it as the optimal solution for airborne applications.

### 3.2. Quick-Look Imaging Function Experiment

To verify that the system has imaging and quick view functions, simulated experiments were conducted under laboratory conditions. The physical prototype shown in Figure 9 was mounted on an optical swing simulator to emulate airborne push-broom imaging scenarios. The optical swing simulator can take the camera system to rotate left and right to simulate the push broom imaging process. The experimental subjects are buildings outside the window, which have significant differences in spectral characteristics. The experiment primarily evaluated the imaging and quick-look functions of the system.

Figure 12 is the schematic diagram of the experimental scheme, and Figure 13 is the picture of the experimental site. Figure 14 presents the quick-look visualization output via HDMI interface.

Figure 15 and Figure 16 shows experimental results of hyperspectral images. Figure 15 presents standard push-broom imaging results capturing building facades, rooftops, glass surfaces, sky, and foliage. Figure 16 displays imagery acquired during oscillatory scanning of building exteriors, exhibiting distinct spatial symmetry due to the bidirectional scanning pattern. Both images employ a three-band composite of adjacent spectral channels (1080 nm, 1098 nm, 1116 nm), resulting in predominantly gray-toned visualizations. The chromatic artifacts visible in the imagery originate from inherent optical characteristics of the short-wave infrared camera system rather than data transmission effects within the processing pipeline.

### 3.3. Real-Time Processing Function Experiment

To validate the real-time processing capability of this system in an airborne environment, this experiment employs relative radiometric correction of hyperspectral images as a representative computational task. The entire process consists of two steps: (1) computing a relative radiometric coefficient matrix by averaging multiple frames of accumulated data; (2) applying element-wise multiplication between the coefficient matrix and the target image to obtain the corrected image.

The image data in this study contains 640 pixels per line with 270 spectral bands. Each frame represents the spectral data of one line. Every 512 frames correspond to a spatial image of 640 × 512 pixels with 270 spectral bands. We first use 5120 frames (equivalent to 10 images) to compute a relative radiometric coefficient matrix of size 640 × 270. Then, we apply this matrix to correct 512 frames (one image) of target data, recording the processing time.

Table 7 presents the recorded processing times, while Figure 17 shows a comparison of images before and after relative radiometric correction. Based on computation time and image processing results, the system demonstrates real-time image processing capabilities and shows potential for implementing more complex real-time processing functionalities.

The timing analysis was conducted on the onboard ARM processor featuring an octa-core CPU with 4 × Cortex-A76 @ 2.4 GHz and 4 × Cortex-A55 @ 1.8 GHz, equipped with 8 GB DDR4 RAM.

## 4. Conclusions

Focused on resolving the fundamental conflict between massive data volume and real-time processing in UAV-borne hyperspectral imaging, this paper presents an Onboard Real-Time Hyperspectral Image Processing System Design for Unmanned Aerial Vehicles based on an FPGA-ARM dual-processor architecture. The integrated system incorporates a shortwave infrared hyperspectral camera, a power module, an IMU navigation sensor, a key control module, an FPGA-ARM dual-processor data acquisition and real-time processing module, and SATA SSD. Through hardware-software co-design and algorithmic optimization, we achieve full-pipeline real-time operation spanning “acquisition, storage, processing, and display” of hyperspectral data. The compact integrated system exhibits a total weight of only 6 kg and power consumption of 40 W, rendering it highly suitable for most onboard applications.

Experimental validation under simulated airborne push-broom scenarios confirms stable system operation, demonstrating rapid terrain imaging capabilities and real-time quick-look visualization functionality. The system sustains a storage rate exceeding 200 frames per second for 640 × 270 pixel imagery; further evidencing its real-time processing competence, it processes nearly 5000 frames per 0.1 s for representative tasks such as relative radiometric correction. These capabilities provide critical technical support for the efficient deployment of hyperspectral technology in dynamic operational environments.

Future work will deploy this device in actual flight missions to validate its performance in real-world scenarios. Flight trials are currently scheduled for September 2026 in Beijing’s suburban airspace. Given the core system weight of 6 kg, it requires mounting on a compatible gimbal, and total weight is about 10 kg. With additional modules including batteries, video, and data transmission systems, the complete payload will approach 15 kg. This necessitates identification of suitable UAV platforms capable of carrying this payload. Furthermore, extensive coordination with airspace authorities and site management is required prior to execution. These prerequisite tasks confirm our planned timeline for conducting actual flight tests in 2026.

## Figures and Tables

**Figure 1 sensors-25-04822-f001:**
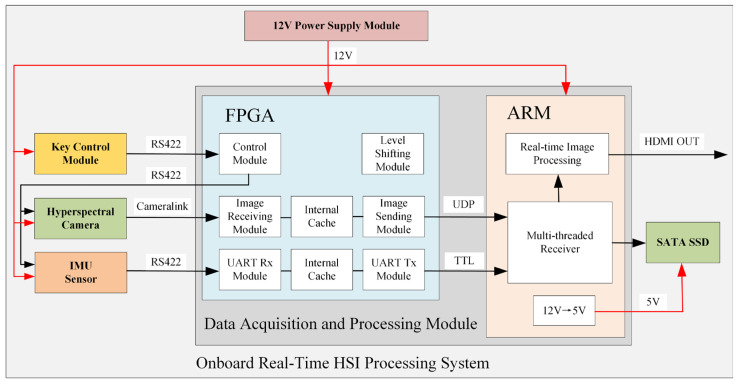
Onboard Real-Time HIS Processing System.

**Figure 2 sensors-25-04822-f002:**
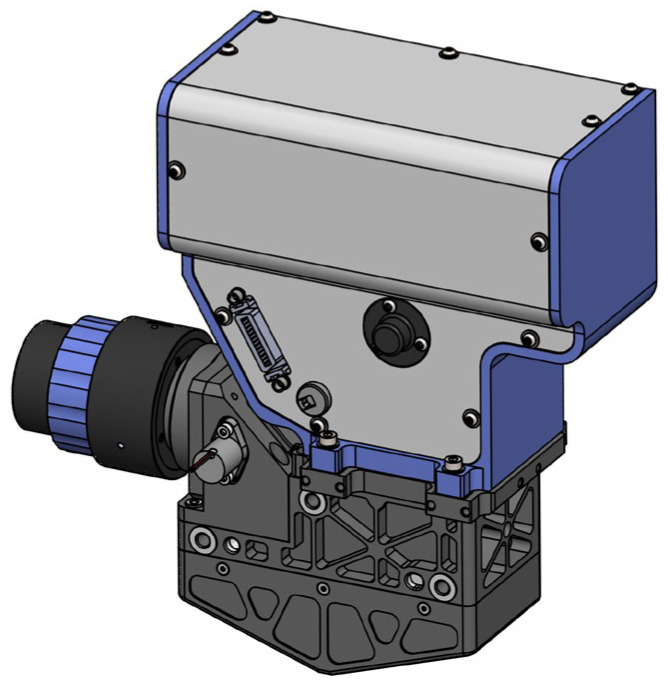
Three-dimensional rendered exterior view of the Micro-Hyperspec SWIR 640 (NBL Imaging System Ltd., Guangzhou, China).

**Figure 3 sensors-25-04822-f003:**
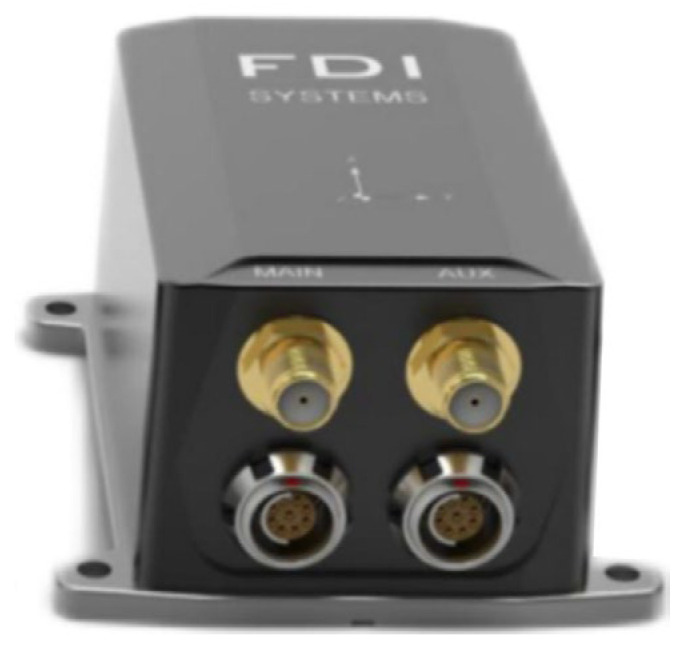
Appearance diagram of the EPSILON D IMU sensor (FDIsystems Ltd., Hefei, China).

**Figure 4 sensors-25-04822-f004:**
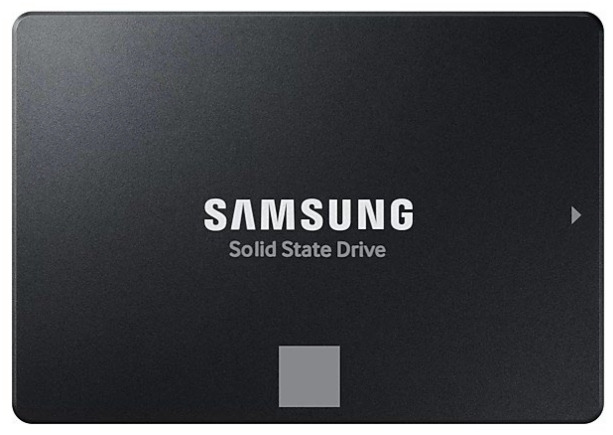
Physical configuration of the SATA SSD (Samsung Electronics Ltd., Seoul, Republic of Korea).

**Figure 5 sensors-25-04822-f005:**
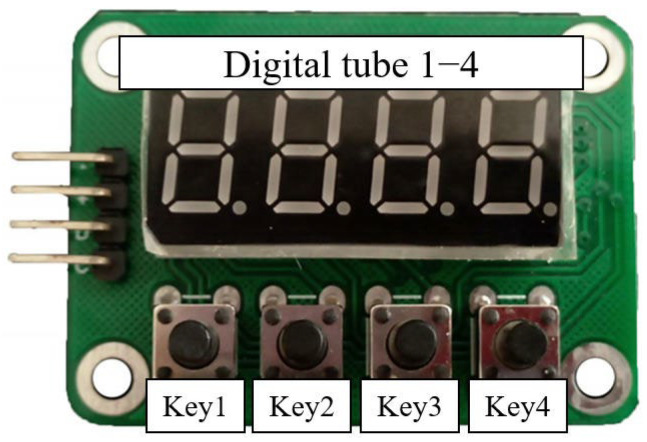
Appearance diagram of key-digit interface panel.

**Figure 6 sensors-25-04822-f006:**
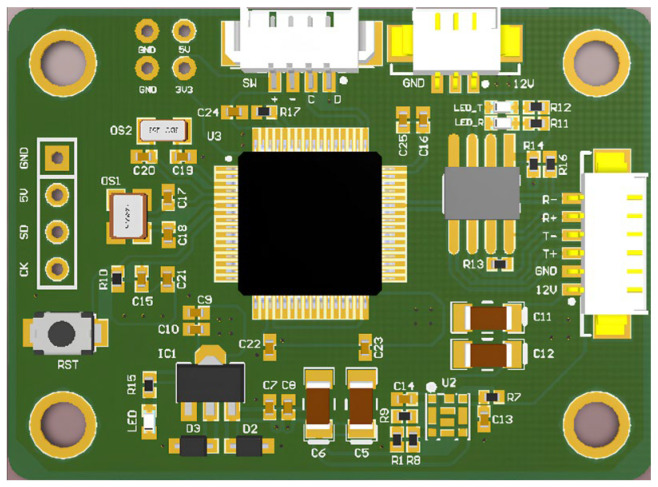
The core control board of key control module.

**Figure 7 sensors-25-04822-f007:**
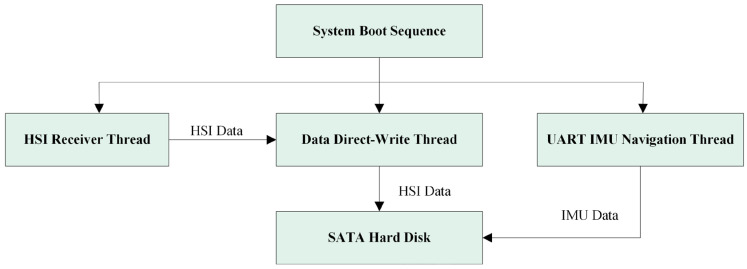
ARM-side three-thread operation mode block diagram.

**Figure 8 sensors-25-04822-f008:**
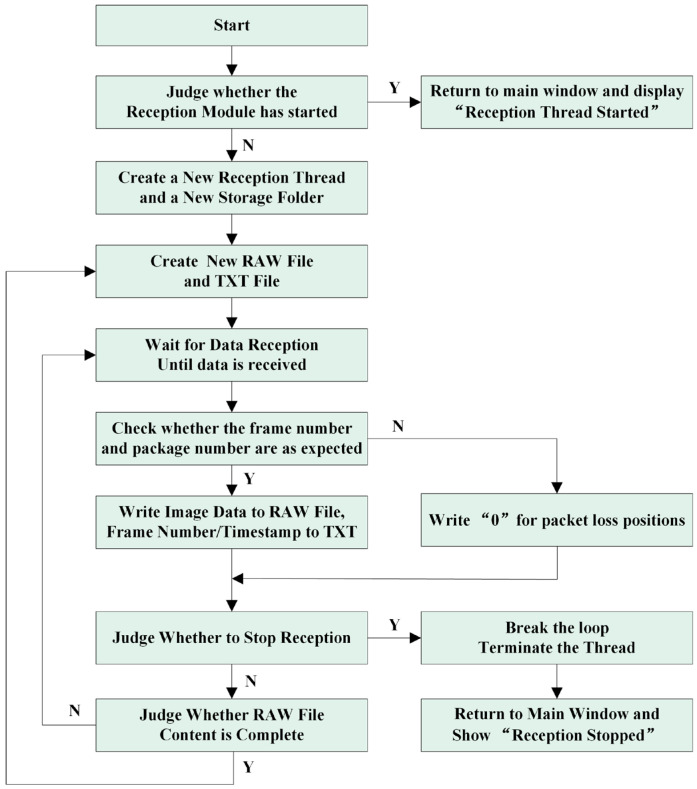
HSI receiver thread block diagram.

**Figure 9 sensors-25-04822-f009:**
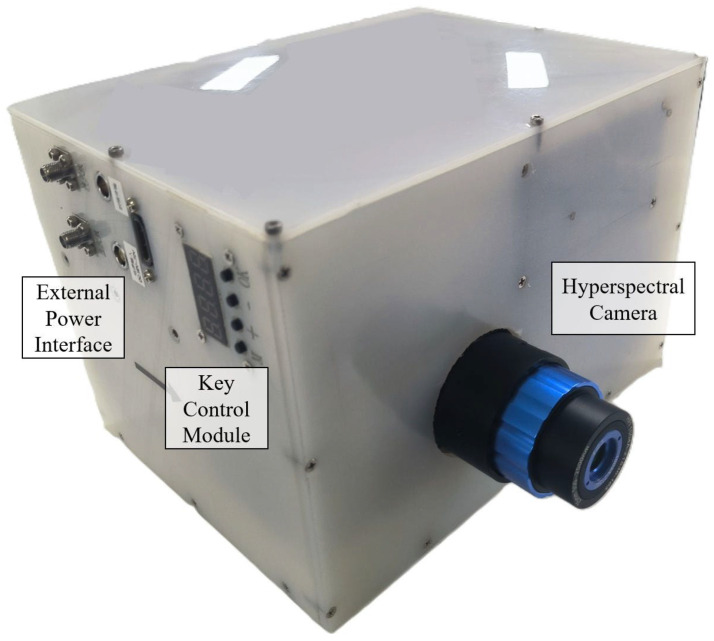
Physical assembly external view of the UAV payload.

**Figure 10 sensors-25-04822-f010:**
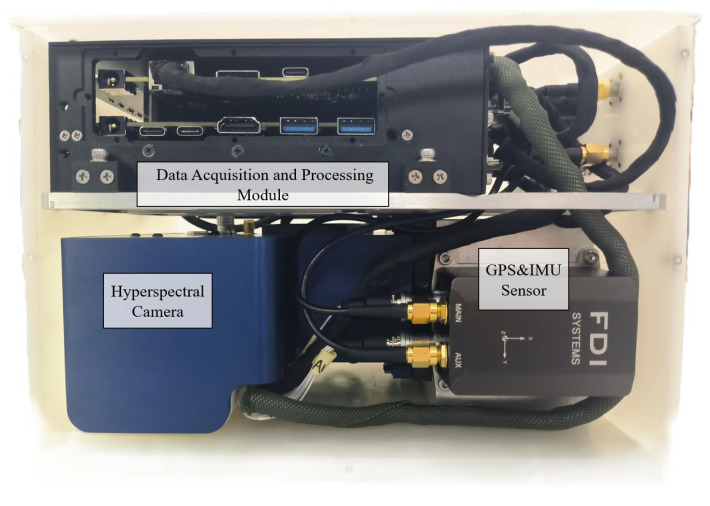
Internal view of dual-layer physical assembly.

**Figure 11 sensors-25-04822-f011:**
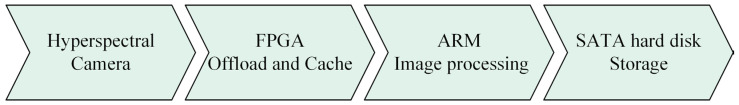
The complete data transmission pipeline of the system.

**Figure 12 sensors-25-04822-f012:**
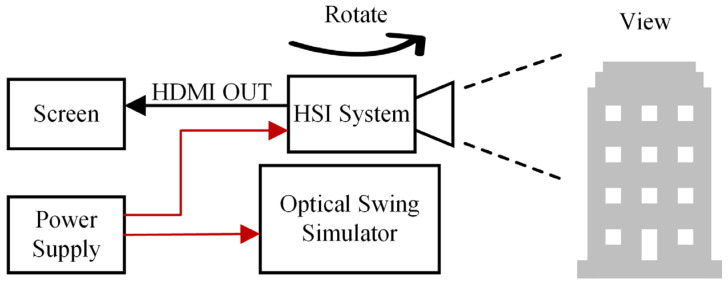
The experimental scheme.

**Figure 13 sensors-25-04822-f013:**
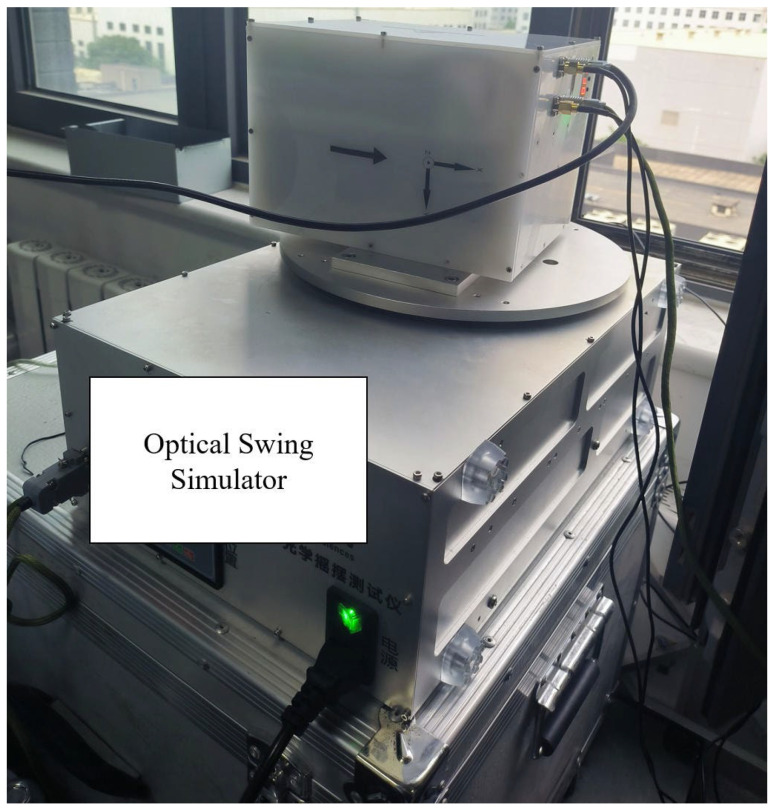
The experimental site.

**Figure 14 sensors-25-04822-f014:**
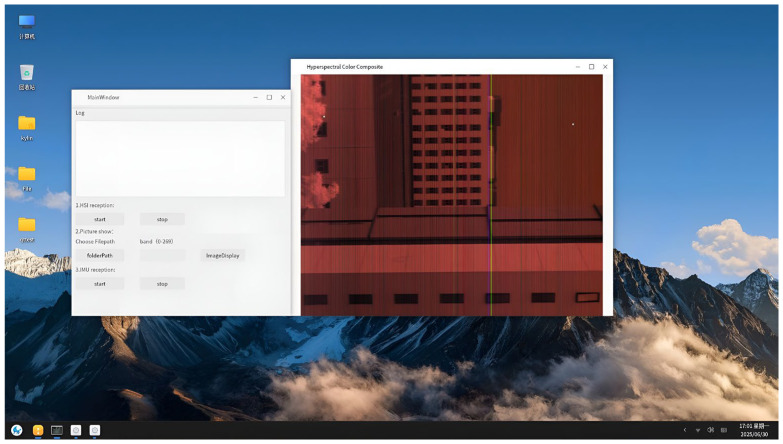
The quick-look visualization output.

**Figure 15 sensors-25-04822-f015:**
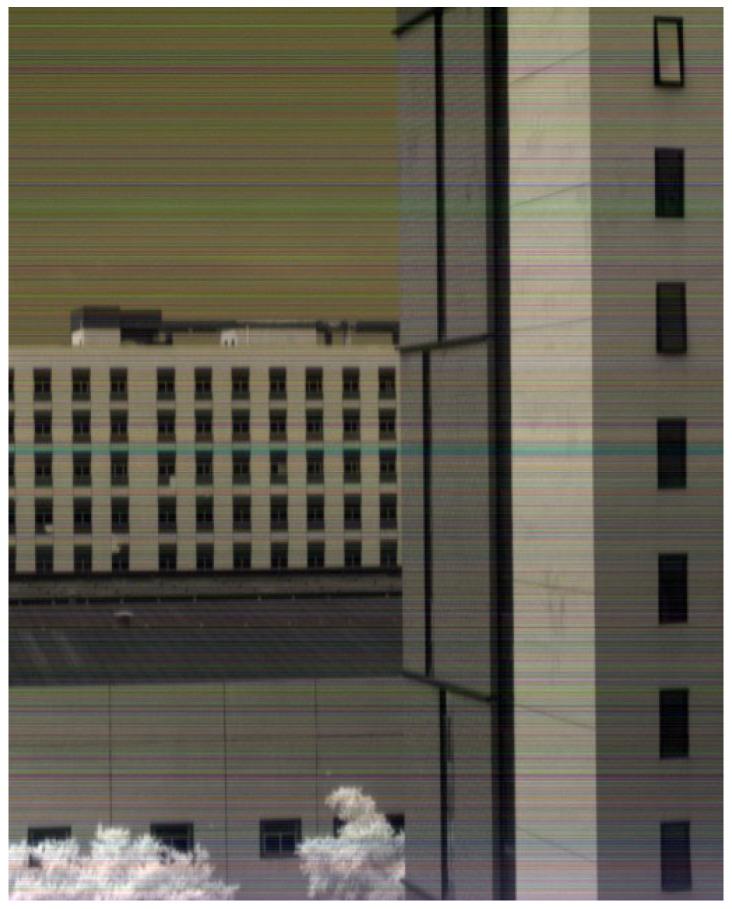
Standard push-broom acquired imagery.

**Figure 16 sensors-25-04822-f016:**
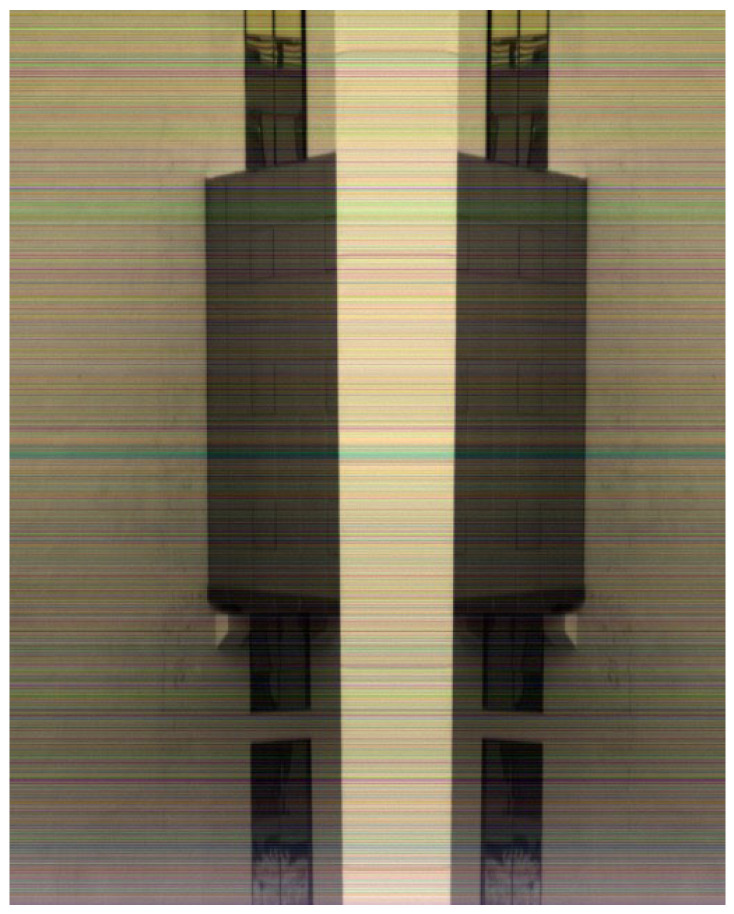
Bidirectional oscillatory push-broom comparison image.

**Figure 17 sensors-25-04822-f017:**
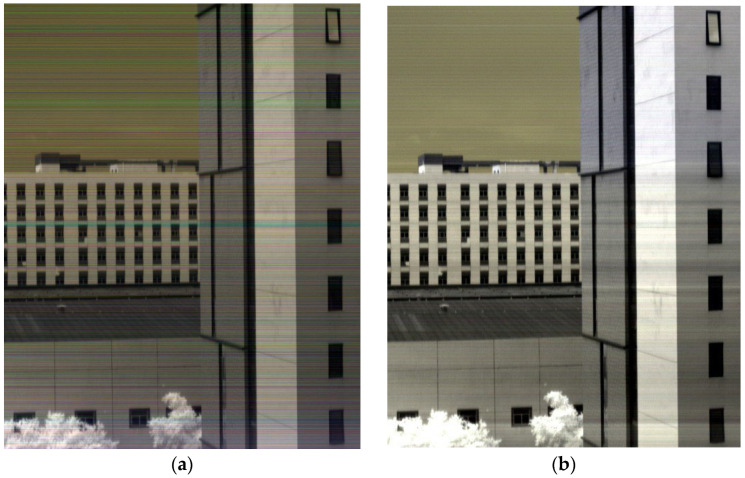
Results of relative radiometric correction. (**a**) Original image; (**b**) corrected image after relative radiometric processing.

**Table 1 sensors-25-04822-t001:** Technical specifications of the Micro-Hyperspec SWIR 640.

Item	Specification
Spectral Range	900–2500 nm
Sensor Type	Mercury Cadmium Telluride (MCT)
Pixel Size	15 μm
Aperture	F/2.5
Slit Length	10.5 mm
Spectral Sampling Value	6 nm/pixel
Slit Width	20 μm
Spectral Channel Number	270
Spatial Channel Number	640
Maximum Frame Rate	>200
A/D Conversion Bit Depth	16 bit
Cooling Method	Stirling Cooling
Camera Data Interface	Base CameraLink
Weight	1.6 kg
Maximum Power Consumption	14 W

**Table 2 sensors-25-04822-t002:** Technical specifications of the EPSILON D IMU sensor.

Inertial Navigation Sensor	Item	Specification
Position Accuracy	Horizontal	Single point: 1 m SBAS: 0.6 m DGPS: 0.4 m RTK: 1 cm
	Vertical	10 cm
Velocity Accuracy	Horizontal/Vertical	Single point: 0.1 m/s RTK: 0.03 m/s
Attitude Accuracy	Roll/Pitch	Static state: 0.05° Dynamics: 0.1°
3-axis Accelerometers	Full Scale Bandwidth Bias stability	−8 g to 8 g 500 Hz <0.4 mg
3-axis Gyroscopes	Full scale Bandwidth Bias stability	−2000°/s to 2000°/s 300 Hz 2°/h *
3-axis Magnetometers	Full scale Bandwidth Bias stability	−800 μT to 800 μT 200 Hz 20 nT
GNSS Receiver	Signal Tracking	2 × 184-channel GPS L1C/A L2C, GLO L1OF L2OF, GAL E1B/CE5b, BDS B1l B2l, QZSS L1C/A L1S L2C, SBAS L1C/A
	Output Frequency	20 HZ
Internal Barometric Altimeter	Resolution	300–1200 hPa
Mechanical	Size Weight	55 × 55 × 36 mm 108 g
Max Power Consumption	-	3 W

* Allan variance standard testing environment.

**Table 3 sensors-25-04822-t003:** Technical specifications of the SATA SSD.

Item	Specification
Model	SAMSUNG 870 EVO
Interface	SATA 6 Gb/s
Form Factor	100.0 × 69.85 × 6.8 mm
Capacity	2 TB
Sequential Read	560 MB/s
Sequential Write	530 MB/s
Weight	46 g
Average Power Consumption	2.5 W
Maximum Power Consumption	4.5 W

**Table 4 sensors-25-04822-t004:** Comparison table of digital tube display meaning.

Digital Tube 1	Digital Tube 2	Digital Tube 3	Digital Tube 4	Meaning
G	-	-	L	Low Gain
	-	-	H	High Gain
P	-	Cycle High Digit 0–9	Cycle Low Digit 0–9	Cycle (Unit: 20 ms)
E	-	Exposure Time High Digit 0–9	Exposure Time Low Digit 0–9	Exposure Time (Unit: 10 ms)

**Table 5 sensors-25-04822-t005:** Technical specifications of the overall system.

Item	Specification
Power Supply	DC 12 V
Form Factor	16.5 × 21.5 × 15 cm
Weight	6 kg
Video interface	HDMI OUT
Operational Temperature	−40–60 °C
Average Power Consumption	40 W

**Table 6 sensors-25-04822-t006:** Record of packet loss in UDP transmission.

Experimental ID	Frame Rate	Total Frames (30 min)	UDP Packet Loss Count	Complete Frame Loss Count	Total Loss Count	Complete Frame Loss Rate
1	50	90,000	0	0	0	0
2			0	0	0	0
3			0	0	0	0
4	100	180,000	1	0	1	0
5			2	1	3	0.06‱
6			1	1	2	0.06‱
7	200	360,000	2	1	3	0.03‱
8			2	2	4	0.06‱
9			3	1	4	0.03‱

**Table 7 sensors-25-04822-t007:** Processing time of relative radiometric correction.

Processing Stage	Data Volume	Operation Complexity	Time (ms)
Coefficient Calculation	5120 frames	640 × 270 × 5120 additions	1044
Image Correction	512 frames	640 × 270 × 512 multiplications	322
Total	5632 frames	Combined operations	1366

## Data Availability

The original contributions presented in this study are included in the article. Further inquiries can be directed to the corresponding author.

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
