# Peer review of "Onboard Real-Time Hyperspectral Image Processing System Design for Unmanned Aerial Vehicles"

_sensors, 2025, doi:10.3390/s25154822_

Round 1
Reviewer 1 Report
Comments and Suggestions for Authors
This manuscript presents a well-structured and timely design of a UAV-compatible hyperspectral image processing system utilizing a dual FPGA-ARM architecture. The paper clearly outlines the system’s architecture, implementation, and experimental validation under simulated conditions. The contribution is relevant, technically sound, and aligns well with the scope of Sensors.
However, the manuscript would benefit from minor improvements in positioning, clarity, and language, as well as a slightly deeper discussion on novelty and comparison with existing works. Therefore, I recommend minor revision before acceptance.
- The system has only been tested in simulated lab conditions. While acceptable at this stage, please add a clear statement in the Conclusion about the timeline or plan for real-world UAV flight testing.
- Line 25 (Abstract): Consider rephrasing “overcome hyperspectral processing bottlenecks” with “address hyperspectral data processing bottlenecks more efficiently”.
- Line 38-44: You may cite a few more specific use cases or real-time UAV scenarios to emphasize practical impact.
- Table 6 shows some excellent data representation. Please specify how much loss (in % or total dropped packets) is tolerable in mission-critical applications, and how your rate compares.
- Please include hardware specs (e.g., CPU clock, RAM size) used for this timing analysis for reproducibility in Table 7.
- Line 49: "…leading to exponential growth in data volume." → "…resulting in substantial data volumes."
- Line 217–220: Consider splitting the following long sentence for readability."The FPGA-ARM dual-processor architecture overcomes single-processor perfor
mance limitations through heterogeneous task partitioning: The FPGA serves as a hard-
ware acceleration engine, dedicated to high-speed hyperspectral data acquisition, buffer
ing, and real-time preprocessing, thereby freeing ARM computational resources. "
Reviewer 2 Report
Comments and Suggestions for Authors
Please see the attachment.
